# Strongyloides and COVID-19: Challenges and Opportunities for Future Research

**DOI:** 10.3390/tropicalmed8020127

**Published:** 2023-02-19

**Authors:** Daniel Seeger, Enrique Cornejo Cisneros, Jose Lucar, Rachel Denyer

**Affiliations:** 1Infectious Diseases Section, DC Veterans Affairs Medical Center, Washington, DC 20422, USA; 2Division of Infectious Diseases, School of Medicine and Health Sciences, George Washington University, Washington, DC 20052, USA; 3Instituto de Medicina Tropical Alexander von Humboldt, Universidad Peruana Cayetano Heredia, Lima, 15102, Peru

**Keywords:** Strongyloides, COVID-19

## Abstract

*Strongyloides stercoralis* is a soil transmitted helminth endemic to tropical and subtropical areas that can persist for decades in immunocompetent human hosts as a chronic asymptomatic infection. The use of corticosteroids, a mainstay of treatment for patients hospitalized with severe coronavirus disease (COVID-19), can trigger a life-threatening Strongyloides hyperinfection syndrome and disseminated disease. We identified 22 previously published cases of strongyloidiasis occurring in individuals with COVID-19, with one death reported among the seven patients who had Strongyloides hyperinfection syndrome. A total of seventeen patients had previously received corticosteroids, and of the five with no prior corticosteroid use, one presented with hyperinfection syndrome. We identify the key challenges in the diagnosis and treatment of Strongyloides within the context of COVID-19, including our imprecise knowledge of the global distribution of Strongyloides, the overlapping symptoms and signs of COVID-19 and Strongyloides hyperinfection syndrome, the limited utility of eosinophilia as a clinical marker for strongyloidiasis in this setting, the lack of validated algorithms to screen for Strongyloides prior to corticosteroid use, and the paucity of treatment options for critically ill patients with COVID-19 who cannot take oral ivermectin. Future research should focus on improved diagnostic methods and population prevalence estimates, optimizing the approaches for Strongyloides screening in persons with COVID-19 (including clinical trial participants and strategies for resource-limited settings) and better defining the role of pre-emptive treatment.

## 1. Introduction

*Strongyloides stercoralis* is a soil-transmitted helminth endemic to tropical and subtropical areas which typically causes an asymptomatic chronic infection [1]. This often goes unrecognized in immunocompetent human hosts and can persist for decades after the initial exposure, but in immunocompromised individuals, *S. stercoralis* can cause a life-threatening hyperinfection syndrome and disseminated disease [1,2,3]. Strongyloides is able to complete its entire life cycle within a single human host, with rhabditiform larvae becoming infective filariform larvae within the human gastrointestinal tract, which can then penetrate the intestinal mucosa or perianal skin of the host, causing autoinfection. This phenomenon allows Strongyloides to maintain low levels of infection within human hosts that can persist for decades in the absence of ongoing exogenous exposure.

Corticosteroid administration is the most commonly recognized cause of Strongyloides hyperinfection syndrome [2,3]. The mechanisms through which corticosteroids drive accelerated autoinfection and hyperinfection remains incompletely understood. The iatrogenic suppression of eosinophils by corticosteroids, primarily through the induction of apoptosis, may predispose patients with a chronic Strongyloides infection to hyperinfection and disseminated disease. It has also been postulated that a metabolite of corticosteroids is structurally similar to 20-hydroxyecdysone, a hormone that promotes the transformation of rhabditiform larvae into filariform larvae [4,5]. This would result in an accelerated Strongyloides autoinfection cycle, leading to increased burden of disease in the lungs and gastrointestinal tract Strongyloides and the potential dissemination of larvae throughout the host, with morbid clinical consequences.

Strongyloides hyperinfection can be provoked by a wide spectrum of corticosteroid doses and durations. A multicenter study of 133 patients diagnosed with Strongyloides hyperinfection found that 83% of the patients had previously been treated with corticosteroids, with a median dose of prednisone of 40 mg per day. The median time from corticosteroid initiation to the development of hyperinfection was 42 days, and it was observed that the corticosteroid dose had been recently increased in 47% of cases [2]. It should be noted, however, that doses as low as 1 mg of dexamethasone, durations as short as six days, and locally injected corticosteroids have all been implicated as causes of Strongyloides hyperinfection [6,7,8].

The COVID-19 pandemic is estimated to have caused 18.2 million excess deaths between 2020 and 2021, with South Asia, North Africa and the Middle East, and Eastern Europe accounting for the largest numbers of excess deaths [9]. A breakthrough in the fight against COVID-19 was made by the RECOVERY trial [10], which demonstrated that treatment with dexamethasone was associated with a statistically significant reduction in the 28-day mortality of persons with COVID-19 who were receiving respiratory support. Subsequently, corticosteroid use in hospitalized patients with severe or life-threatening COVID-19 and respiratory failure was rapidly endorsed by multiple treatment guidelines, including those from the World Health Organization (WHO) and the U.S. National Institutes for Health (NIH) [11,12]. However, the widespread use of corticosteroids for the treatment of COVID-19 potentially places persons with undiagnosed chronic *Strongyloides stercoralis* infection at risk for life-threatening hyperinfection syndrome and disseminated disease [13]. Notably, in 2020, more than 2.6 billion people around the world were estimated to be at risk of Strongyloides infection, with the greatest burden of disease disproportionately impacting persons in low- and middle-income country (LMIC) settings [14].

We summarize previous reports of strongyloidiasis occurring in the setting of COVID-19 and evaluate the proposed screening strategies to identify patients at risk for strongyloidiasis in both high and low prevalence settings. We identify the challenges related to the screening, diagnosis, and treatment of strongyloidiasis in the context of the COVID-19 pandemic, highlighting knowledge gaps and proposing specific strategies to address these in future research.

## 2. Previously Reported Cases of Strongyloides Co-infection in Persons with COVID-19

We searched the electronic databases PubMed and Google Scholar from inception to 22 November 2022, using the keywords “COVID-19” or “SARS-CoV-2”, and “Strongyloides” or “strongyloidiasis”. All case reports with the diagnosis of both COVID-19 and any form of strongyloidiasis (chronic or hyperinfection syndrome) were included. Additionally, the reference list from the included studies was used to identify additional relevant literature. We identified 15 manuscripts reporting 22 cases of strongyloidiasis occurring in individuals with COVID-19, of which 7 patients had Strongyloides hyperinfection and 15 had chronic strongyloidiasis. Both asymptomatic and symptomatic patients were included in those with chronic infection. Only one death due to hyperinfection was reported, which is concerning for publication bias, as the anticipated mortality rate for Strongyloides hyperinfection is estimated to be between 60–85% [1,2]. The key demographic and clinical characteristics of the previously reported cases of strongyloidiasis in persons with COVID-19 are summarized in Table 1. 

## 3. Risk Factors for SHS in COVID-19

### 3.1. Epidemiologic Risk

The worldwide prevalence of strongyloidiasis was previously estimated to be between 30–100 million, however, a recent study using a spatiotemporal statistical modeling approach estimated a global prevalence of 8.1%, corresponding to 613.9 million people infected worldwide [30]. This underestimation by a factor of around 10 likely relates to the limitations in the available diagnostic testing for Strongyloides infections, and the understudied nature of this neglected tropical disease. Although data is limited, infection rates in sub-Saharan Africa, obtained through community-based surveys, are estimated to be as high as 90% in some regions [31]. In Southeast Asia, prevalence estimates range from 17% in Thailand to 33% in Laos [32,33].

There are also localized areas of endemicity documented in more temperate climates. In the United States, Strongyloides is considered endemic in rural parts of the southeast and Appalachia, where the seroprevalence of strongyloidiasis is estimated to be about 2% [34,35]. Our knowledge of the geographic distribution of Strongyloides species globally is limited, even in high-income country (HIC) settings. For example, recent modeling suggests far larger areas in the United States provide a suitable habitat for Strongyloides than previously described [36], consistent with previous reports of strongyloidiasis without travel to areas of higher endemicity [37]. A recent study performed in rural Alabama found a 7.3% prevalence of *S. stercoralis* using stool PCR analysis [38], which is higher than that of previous studies which used less sensitive stool microscopy detection methods [39].

Although predominantly endemic in tropical and sub-tropical regions, where hot and humid climates facilitate its transmission through contaminated soil, the global migration patterns and longstanding duration of the asymptomatic Strongyloides infection have facilitated a wider geographic distribution of cases. Multiple studies describe a significant burden of disease in international migrants living in low endemicity and HIC settings. For example, a study from referral centers in metropolitan Barcelona, Spain, found that over 90% of those diagnosed with strongyloidiasis were migrants, with a median length of residence in Spain of 4 years [40]. A recent systematic review and meta-analysis of migrants born in endemic countries and arriving or living in countries of low endemicity found a pooled seroprevalence of 17.3% in migrants from East Asia and the Pacific, 14.6% in those from sub-Saharan Africa, and 11.4% in those from Latin America and the Caribbean [41]. Sudanese and Somali refugees that had relocated to the United States were found to have seroprevalence rates of 46% and 23%, respectively [42]. 

Furthermore, migrant populations have also been shown to be disproportionately impacted by COVID-19. A study from Norway found that 35–50% of the confirmed cases of COVID-19 were in migrants, despite them only making up about 15% of the general population. [43]. Migrant workers are overrepresented in lower paid jobs and crowded living situations, and as such, are more vulnerable to the spread of COVID-19 [44]. As such, migrant populations appear to be at an increased risk for both COVID-19 and Strongyloides infection.

In addition to geographic risks, previous studies have also reported a continued increased prevalence of strongyloidiasis throughout adult life [45]. This places older patients, already at a high risk for poor outcomes from COVID-19, also at an increased risk for chronic strongyloidiasis. Pregnant women are an important special population that are known to be at an increased risk for both parasitic infection and worse outcomes from COVID-19 [46,47].

In the setting of the global COVID-19 pandemic, where large populations of individuals worldwide are being treated with corticosteroids, the uncertainty surrounding the estimates of Strongyloides prevalence is problematic. There is an urgent need to better understand the geographic distribution and risk factors associated with Strongyloides infection, in order to more accurately predict which individuals are at an increased risk for hyperinfection syndrome and disseminated disease in the setting of COVID-19. 

Mathematical modeling approaches, including the use of machine learning and artificial intelligence approaches, could be used to identify the communities at risk for Strongyloides which may not have been directly sampled, and to provide better estimates of prevalence in the context of limited sampling data from the population affected. We identified only two previous studies which applied machine learning approaches to Strongyloides, neither of which pertained to COVID-19. The first applied an unsupervised machine learning technique to a clustering of *S. stercoralis* and *S. fuelleborni* genotypes in humans and dogs, while the other used a supervised machine learning model to demonstrate an association between the presence of gastrointestinal parasites (including Strongyloides) and the gut microbiome composition of a Cameroonian population [48,49]. However, the uncertainties that can lead to imprecision or inaccuracy within such mathematical models of soil-transmitted helminth infections should be carefully delineated, considered, and estimated, with a previous meta-analysis highlighting deficiencies in this regard in the studies that they analyzed [50]. The sources of uncertainty impacting Strongyloides epidemiologic modeling include not only imperfections in the sampling methods and the limitations of the diagnostic testing used, which can cause incorrect disease state assignments, but also epistemic biological factors such as the heterogeneity of detectability of the Strongyloides in different hosts, variations in the distribution and intensity of Strongyloides organisms among host populations, and the impact of host demographic factors.

### 3.2. Immunosuppression

Following the initial results of the RECOVERY trial published in June 2020, corticosteroid therapy has become a standard of care for hospitalized patients with severe COVID-19 requiring respiratory support across the globe [11]. Based on the association between corticosteroids and accelerated Strongyloides autoinfection, the most utilized corticosteroid regimen of 6 mg of dexamethasone once daily for up to 10 days puts individuals with COVID-19 at risk for Strongyloides hyperinfection and disseminated disease. 

IL-6 inhibitors, such as tocilizumab, have also been used to treat certain hospitalized patients with severe COVID-19, primarily based on the findings of the RECOVERY and REMAP-CAP trials [51,52]. An analysis of the safety data from phase 3 trials of tocilizumab use for rheumatoid arthritis showed an increased risk of bacterial infections, as well as opportunistic infections with pathogens such as mycobacteria, *Pneumocystis*, *Candida,* and *Cryptococcus* [53,54]. More frequent or severe parasitic infections were not identified in those studies. Anakinra, a recombinant IL-1 receptor antagonist, was recently authorized for use in the United States and Europe in certain patients with severe COVID-19, primarily based on the findings of the SAVE-MORE trial [55]. Parasitic infections have only rarely been reported in individuals receiving IL-1-targeted agents. For example, a case of visceral leishmaniasis after 6 months of treatment with anakinra for juvenile idiopathic arthritis has been reported [56]. Janus kinase (JAK) inhibitors, such as baricitinib, are also used in certain patients with severe COVID-19, primarily based on the findings of the RECOVERY [57] and COV-BARRIER trials [58]. We were unable to find any previous association between the use of JAK inhibitors and an increased risk of helminth infections.

There have been published reports of Strongyloides hyperinfection in patients who received tocilizumab [17,22,24], baricitinib [15], and anakinra [24,27] for the treatment of severe COVID-19. However, as is recommended by major treatment guidelines, all of those patients also received corticosteroids during their course of treatment. As such, the role played by the second immunomodulatory agent is uncertain. Interestingly, there is a single case report of Strongyloides hyperinfection in an individual who did not receive corticosteroids or other immunomodulatory agents [19], which indicatesthat other factors could play a role beyond the immunomodulatory therapy received. 

## 4. Diagnosis

### 4.1. Clinical and Radiologic Features

The diagnosis of Strongyloides hyperinfection in an individual with COVID-19 remains a clinical challenge and requires a high index of suspicion. There is a large degree of clinical overlap in the symptoms and signs of severe COVID-19, driven primarily by acute respiratory distress syndrome and multisystem organ damage, and Strongyloides hyperinfection, manifested by an increased larval burden and migration through the respiratory system, gastrointestinal tract, and skin. Thus, the diagnosis of Strongyloides hyperinfection may be overlooked amid the complexity of a severe case of COVID-19. For example, diarrhea and abdominal pain are relatively common in COVID-19, occurring in up to 20% of patients, whereas gastrointestinal symptoms are reported in more than 70% of those with Strongyloides hyperinfection [2,59]. Up to 70% of those with COVID-19 may experience fever, cough, or shortness of breath, while more than 80% of those with Strongyloides hyperinfection may experience respiratory symptoms and fever [2,59]. *Larva currens* is considered to be pathognomonic for strongyloidiasis, but urticaria has also been associated with both chronic Strongyloides infection and COVID-19 [60,61,62,63].

The common radiologic findings for both COVID-19 and Strongyloides hyperinfection are pulmonary ground-glass opacities and septal thickening [64,65]. Less frequently, migratory pulmonary opacities have been described for both conditions [64,66,67]. However, the association of COVID-19 with migratory pulmonary opacities has been anecdotal, occurring in the context of cryptogenic organizing pneumonia and before the widespread use of corticosteroids [68,69]. 

Both COVID-19 and Strongyloides hyperinfection can be complicated by secondary Gram-negative bacterial infections. With Strongyloides, the increased larval migration through the walls of the gastrointestinal tract has been associated with infections with enteric bacterial flora, including cases of polymicrobial infection. These infections can manifest as meningitis, pneumonia, or sepsis. On the other hand, COVID-19 can also be complicated by pneumonia and sepsis, predominantly as a complication of mechanical ventilation. A systematic review and meta-analysis of published studies estimated that between 20–30% of COVID-19 patients have secondary infections [70]. The most commonly identified bacteria were *Acinetobacter* spp. (22.0%), *Pseudomonas* (10.8%), and *Escherichia coli* (6.9%), which may obscure the classic association of Gram-negative bacterial infections with Strongyloides hyperinfection.

### 4.2. Eosinophilia

The presence of eosinophilia is generally considered a useful indicator of chronic Strongyloides infection, and it may be present in up to 80% of patients at the time of diagnosis [71]. However, during Strongyloides hyperinfection, eosinophilia is often absent, with a prevalence of 34% in a single, large retrospective study [1,2,71]. The absence of eosinophilia in cases of Strongyloides hyperinfection may be at least in part due to corticosteroid therapy [72]. Additionally, most patients with COVID-19 (51.9–81.2%) have eosinopenia at the time of presentation, which may be associated with worse outcomes [73]. Eosinopenia appears to be more common in SARS-CoV-2 infection than influenza [74], and in one study conducted during the early part of the pandemic, eosinopenia was found to be the best single biomarker for predicting a diagnosis of COVID-19, with a sensitivity of 75% and specificity of 69% [75]. In contrast, increasing eosinophil counts have been associated with the recovery from COVID-19 [76,77].

The strong association between SARS-CoV-2 infection and eosinophil suppression may lead to a reduced sensitivity of eosinophilia as a marker for chronic Strongyloides infection in the setting of COVID-19. For example, one published case report described a 59-year-old Ecuadorian male residing in Belgium who had chronic eosinophilia, but developed eosinopenia at the time of his admission to hospital with COVID-19 (prior to the initiation of corticosteroids), and had a return of eosinophilia during his recovery phase after completing a course of corticosteroids [27]. Eosinophil suppression due to COVID-19 itself may have played a role in the case of Strongyloides hyperinfection in an individual with COVID-19 that did not receive corticosteroids or other immunomodulating agents [19]. It should be noted that there are also cases of eosinophilia in the setting of COVID-19 due to adverse drug reactions such as DRESS syndrome [78] and vasculitis [79]. Overall, eosinophilia appears to be an unreliable marker for strongyloidiasis in the setting of COVID-19, although further research is needed. 

### 4.3. Laboratory Techniques for Strongyloides Diagnosis

Even in the absence of concomitant COVID-19, the diagnosis of Strongyloides remains a clinical challenge. Traditionally, stool microscopy has been used for the direct visualization of larvae, but its utility is limited by low sensitivity due to the intermittent excretion of larvae during the chronic stage of infection. It has been estimated that it may take up to seven stool samples to achieve diagnostic certainty when relying solely on stool microscopy [80]. Additional stool-based studies, such as the Baermann stool concentration method and agar plate culture techniques, increase the diagnostic yield for a single stool specimen and have been successfully implemented in several endemic settings [81], but can be a technical challenge due to their time-consuming nature. A Strongyloides polymerase chain reaction (PCR) in stool samples is more specific than other stool-based studies, but is not widely available yet, and its implementation is often not feasible in LMIC healthcare settings with limited resources such as equipment, reagents, and trained laboratory personnel [82]. While the modified Baermann method combined with Koga agar plate culture has been shown to be as sensitive as PCR, it has a longer turnaround time due to the need for culture, which poses challenges in its clinical application in time-sensitive scenarios such as the decision to start corticosteroids for COVID-19 in a patient from a Strongyloides endemic region [33].

Strongyloides serologic testing can also be done in conjunction with stool-based testing. Furthermore, serology has a higher sensitivity compared to stool microscopy if used as a standalone screening test. A systematic review and meta-analysis of strongyloidiasis prevalence among migrants born in endemic countries showed that the Strongyloides seroprevalence was seven-fold higher than the stool-based prevalence [41]. However, the limitations of serologic testing include its cross-reactivity with other soil-transmitted helminths and an inability to distinguish between current and past infection. The sensitivity of these tests is also lower in immunocompromised individuals, including those receiving corticosteroids [3], with a reduced sensitivity from 96% in immunocompetent hosts compared to 43% in immunocompromised hosts [83]. It remains unclear if the immune dysregulation associated with acute COVID-19 could affect the sensitivity of serologic assays, which is in itself difficult to study, given the absence of a highly sensitive and specific ‘gold standard’ diagnostic test to use as a comparator. 

Individuals with Strongyloides hyperinfection typically have a large parasite burden in comparison to the very small numbers of parasites shed into the gastrointestinal tract during chronic low-level infection. In hyperinfection, microscopy from sputum, bronchoalveolar lavage (BAL), or gastric lavage samples can readily identify the larvae and confirm the diagnosis [1]. However, the diagnosis of Strongyloides in the setting of severe COVID-19, even in those with hyperinfection, can be extremely challenging in practice. Whilst serologic testing for Strongyloides may be less sensitive in immunocompromised individuals [84] it is also of a lower utility in settings highly endemic for Strongyloides, where a larger share of persons with COVID-19 would be expected to have prior exposure to the parasite. Alternative diagnostic strategies include stool microscopy, which usually requires multiple specimens to increase the diagnostic yield, and BAL, which can be difficult to perform in non-intubated patients with severe COVID-19 due to its high oxygen requirements. Additionally, conducting an aerosol-generating procedure such as BAL in a person with COVID-19 has infection prevention implications and will necessitate additional PPE usage to mitigate the exposure of healthcare workers to SARS-CoV-2, which could be challenging in LMIC settings where N95 masks or powered air-purifying respirators (PAPRs) may not be routinely available, or be of limited supply.

## 5. Screening

The assessment of an individual’s risk for chronic Strongyloides infection should be performed in all patients prior to the initiation of immunomodulating agents, including those with COVID-19 prior to the receipt of corticosteroids. This need is likely underappreciated in low endemicity areas where the primary burden of disease is among the foreign-born population and medical providers may be unaware of the risk of Strongyloides hyperinfection triggered by corticosteroid use [85]. A survey of 363 physicians-in-training, conducted before the COVID-19 pandemic, found that, when presented with a hypothetical case scenario of a person with wheezing and eosinophilia (absolute 900 eosinophils/μL), only 9% of the trainees from the United States could identify the need to evaluate for a parasite, and moreover, 23% incorrectly recommended an empiric course of corticosteroids [86]. Additionally, the screening of the family members and household contacts of the individuals with strongyloidiasis is even more important if the index case also has COVID-19, as these close contacts are likely to be at risk for both infections, and may ultimately receive corticosteroids for COVID-19.

### 5.1. Proposed Screening Protocols

A number of clinical pathways have been proposed for the screening and treating of strongyloidiasis prior to the initiation of corticosteroids specifically for COVID-19. Stauffer and colleagues proposed a screening/treatment algorithm based on the geographic risk of exposure to/infection with Strongyloides and the risk factors for disseminated infection in patients with COVID-19 [87]. This approach was based on the risk categories previously described in the 2016 Committee to Advise on Tropical Medicine and Travel for Canada (CATMAT) [88]. De Wilton and colleagues proposed a broader screening and treating algorithm, taking into account both environmental exposure history and travel history in the assessment of the risk of Strongyloides infection at baseline [89]. Importantly, the algorithm by De Wilton et al. also highlights the symptoms and signs of Strongyloides hyperinfection, such as rash, diarrhea, and respiratory deterioration with immunosuppression, which may help identify those who may have been considered to be low risk at the time of admission. Mohareb and colleagues proposed a more comprehensive screening algorithm to prevent the infectious complications of immunomodulation in foreign-born patients with COVID-19, including latent TB, hepatitis B, Chagas disease, endemic mycoses, herpesviruses, and opportunistic fungal pathogens, as well as the presumptive treatment with ivermectin for Strongyloides, unless a rapid turnaround time serologic assay is available [90]. However, the approaches mentioned above are rather specific to the context in which they were created, and are not achievable in settings that are highly endemic for Strongyloides. 

To address the narrow target populations and geographic applicability of other algorithms, Carnino et al. [91] used a global reference map to define endemicity and combined geographic exposure with the individual’s immune status (awaiting immunosuppression vs. immunosuppressed) to determine the screening methods used. It is important to note that all of the previously described algorithms rely heavily on estimates of the geographic burden of Strongyloides infection in different countries, which, as discussed previously, have limitations due to underreporting. 

It should be noted that in both HIC and LMIC settings, the decision regarding the initiation of corticosteroids for patients with severe COVID-19, thus requiring respiratory support, is a key management priority. The slow turnaround times for Strongyloides diagnostic testing, such as serologic assays at a reference laboratory, represent a significant barrier to the implementation of the proposed screening algorithms [85,90]. Furthermore, hospitalized patients with COVID-19 are often critically ill, which may preclude the ability to gather a full epidemiologic risk profile. This may lead to an underestimation of the pre-test probability for Strongyloides in low endemicity settings, where the risk derives from the exposure and travel history of an individual patient. 

Some authors from higher endemicity settings have highlighted the unavailability of serologic assays, agar plate culture, and molecular methods such as PCR in LMIC healthcare settings as a significant barrier to the implementation of the aforementioned screening algorithms [5]. Additionally, the proposed serology-based Strongyloides screening method for patients with COVID-19 is not as suitable in LMICs and high endemicity settings, where serology is not only often unavailable, but also cannot distinguish between prior and current infection, and can have false positive results due to its cross-reactivity with other endemic parasitic infections [92]. It has been proposed that, for these reasons, and in order to confirm current infection, formol-ether concentration techniques from stool specimens may be a better alternative in LMIC settings with a higher endemicity [92]. 

### 5.2. Previous Studies Screening for Strongyloides in the Setting of COVID-19

A retrospective study done in New York State analyzed the medical records of 106,132 foreign born patients and found 6412 migrants who were diagnosed with COVID-19 [93]. Of the subset with COVID-19, 1% had a history of peripheral eosinophilia and 0.3% had a positive Strongyloides serology. Unfortunately, although 885 of the total patient population born outside the U.S. were noted to have eosinophilia, only 356 patients (40.2%) underwent serologic testing for Strongyloides, highlighting the lack of healthcare provider awareness of this condition in low endemicity settings. A similar study conducted in Spain on 2567 patients with COVID-19 identified 86 patients from geographic areas considered endemic for Strongyloides, and identified 7 patients (8%) with Strongyloides infection based on serology (1 from Africa and 6 from Latin America) [29]. All of these patients survived with good outcomes and none required ICU care. In both studies, however, the authors provided no additional details regarding the treatment for Strongyloides. 

Another study from Spain conducted in mid-2022 used the survey data from 121 hospitals to retrospectively analyze individual hospitals’ diagnostic approach to Strongyloides screening [94]. The authors found that 82% of the respondents did not utilize a standardized screening protocol for patients with COVID-19. In hospitals that did screen for Strongyloides, the approach varied from screening all the patients from endemic regions to screening only the immunocompromised individuals from those areas. Serology was the most common diagnostic test used. Hospitals that used a standardized approach were able to identify 227 cases of Strongyloides infection, 4 of which developed hyperinfection. These findings highlight the real-world evidence of a broad patient population at risk for COVID-19 and Strongyloides co-infection. 

We identified only two studies of prospective screening for Strongyloides in the setting of COVID-19. A study conducted in Ethiopia screened for parasitic infections (including Strongyloides) in the setting of COVID-19, finding that 38% of 751 patients with PCR-confirmed SARS-CoV-2 infection had a parasitic co-infection based on stool microscopy, but no cases of Strongyloides were identified [95]. A separate publication retrospectively reported the outcomes of a prospective screening protocol used for 100 individuals with COVID-19 in Italy [96]. Their approach involved screening for multiple infectious diseases, including strongyloidiasis, in migrants from endemic countries and Italian-born individuals over the age of 65 years. Both serology and agar plate cultures were used for screening. In total, 1 of 54 patients screened positive for Strongyloides via serology and no patients had positive agar plate culture results.

## 6. Treatment of Strongyloides in COVID-19

### 6.1. Presumptive Treatment as an Alternative to Diagnostic Screening Tests

Given the limitations of the screening algorithms discussed above, some authors have suggested the presumptive treatment of chronic strongyloidiasis with ivermectin prior to the initiation of corticosteroids as a reasonable approach to prevent Strongyloides hyperinfection. This is especially pragmatic in low endemicity settings where there may be significant delays in obtaining the laboratory results for high-risk, foreign-born populations [3,85,87,90]. However, this approach could also foster shortages of ivermectin, similar to the shortages observed during the periods of high utilization of off-label ivermectin as an unproven COVID-19 treatment [85]. Prior to empiric treatment, particular attention is required for the individuals from endemic regions for *Loa loa*, given the risk of encephalopathy associated with ivermectin use in high-burden microfilarial infections [97]. 

A total of one to two doses of ivermectin are usually recommended for presumptive Strongyloides treatment prior to immunosuppression. A single dose of ivermectin has been shown to be safe and effective in a randomized controlled trial comparing a single dose of 200 μg/kg to multiple doses [98]. No studies have prospectively evaluated the role of presumptive ivermectin in COVID-19 or the cost-effectiveness of this approach compared to alternative screening algorithms. A retrospective study from Spain assessed patients with COVID-19 from Strongyloides endemic areas who received preventive treatment with ivermectin for two consecutive days [99]. This study found that among 35 patients who were treated preventively, only 3 patients were found to have positive Strongyloides serology and none developed hyperinfection or disseminated disease. Importantly, the intervention was found to be well tolerated and safe for all those treated.

### 6.2. Treatment of Strongyloidiasis and Hyperinfection Syndrome in the Setting of COVID-19

For the treatment of Strongyloides hyperinfection, prolonged courses of ivermectin for at least 10–14 days are often used, and treatment is continued until the daily stool microscopy is repeatedly negative for Strongyloides larvae for at least two weeks [1,100]. All 15 of the previously reported cases of strongyloidiasis in COVID-19 for which treatment regimen is known were treated with ivermectin, with six individuals also receiving albendazole, a drug considered to be a second-line treatment for Strongyloides due to its being less effective than ivermectin if given as a single agent [97,100]. 

Approximately 2% of those with COVID-19 require admission to an intensive care unit, and in the setting of critical illness, there may be a difficulty in administering oral formulations of ivermectin due to malabsorption or shock [59]. Furthermore, the absorption and bioavailability of ivermectin is improved when taken with food, which is not feasible for many patients with respiratory failure due to COVID-19 who are receiving respiratory support [101]. Unfortunately, only oral formulations of ivermectin are licensed for human use, and while subcutaneous and rectal ivermectin have been used previously, the published evidence is merely anecdotal and the optimal dosing via these routes is unknown [97,100,101].

### 6.3. Ivermectin Use in COVID-19 Clinical Trial Settings

Ivermectin has been investigated in multiple studies as a possible treatment for COVID-19. However, it has not been shown to be effective in high-quality trials; therefore, it is not recommended for the treatment of COVID-19. Interestingly, a meta-analysis of 12 randomized clinical trials that evaluated ivermectin use for COVID-19 showed that favorable mortality results were limited to the regions with a high Strongyloides prevalence [102]. Further analysis showed an inverse relationship between the mortality and prevalence of Strongyloides, with a reduction in the relative risk of mortality of 39% for every 5% increase in the Strongyloides prevalence. The observed survival benefit associated with ivermectin for the treatment of COVID-19 in populations with a high Strongyloides prevalence may reflect the benefits of the inadvertent treatment of chronic strongyloidiasis and the prevention of corticosteroid-related hyperinfection, rather than any direct effect of the drug on the SARS-CoV-2 or COVID-19 illness severity. 

In a similar fashion, many countries in Latin America, a region with a high Strongyloides prevalence, implemented the massive use of ivermectin as treatment for COVID-19 despite the lack of evidence showing effectiveness [103]. The large-scale use of ivermectin for COVID-19 may have inadvertently treated many patients with chronic Strongyloides and prevented cases of hyperinfection. Of note, the majority of published cases of strongyloidiasis and hyperinfection associated with COVID-19 that we reviewed were diagnosed in countries with low prevalence for Strongyloides. None of the reported cases diagnosed in countries with a high prevalence for Strongyloides occurred in Latin America. Further studies are needed to confirm whether universal treatment with ivermectin in settings with high Strongyloides endemicity can confer a survival benefit for patients with COVID-19 when compared to other screening and treatment algorithms, such as those proposed by Stauffer and De Wilton [87,89].

## 7. Impact of Strongyloides on COVID-19

Helminth infections provoke a Th-2 predominant immunologic response, and the complex interactions between the parasite and the host allow for the persistence of chronic subclinical infections. Although this balance often leads to minimal symptoms, it is well established that there is a biological toll on the host immune response [104]. Interestingly, the implication of these complex interactions remains ill-defined, and both positive and negative clinical consequences appear plausible, depending on the specific response being studied.

In the context of severe COVID-19, mortality appears to be proportional to the magnitude of the inflammatory response to the initial viral infection [105]. The immune modulation by chronic helminth infections has been proposed to have an overall systemic anti-inflammatory effect on the host [106]. It has been suggested that the decreased severity of COVID-19 observed in Africa may be related to the high prevalence of helminth co-infections [107]. A prospective study of 751 patients diagnosed with COVID-19 in Ethiopia and screened for parasitic infections, including *Strongyloides stercoralis*, using stool microscopy and the modified Ritchie method, found that 11% of those with severe COVID-19 and 52% of those with non-severe COVID-19 were co-infected with intestinal parasites, with parasitic co-infection being associated with significantly lower odds of severe COVID-19 [95]. However, none of the participants in this study were identified as having Strongyloides infection and none received ivermectin. These findings highlight the need for further study on the observed inverse correlation between the incidence of COVID-19 and the burden of helminth infections, including Strongyloides [107]. 

Conversely, helminth infections have been linked to an attenuated immune response to vaccinations and a relative state of anergy. For example, individuals with *Wuchereria bancrofti* infection have a decreased immune response to the tetanus toxoid following tetanus vaccination compared to controls, with a similar pattern described in other helminth infections and vaccinations as well [46,108,109]. Furthermore, a recent phase 1 study of a Plasmodium falciparum vaccine candidate noted a significantly reduced vaccine-induced immunoglobulin G production in participants with a *S. stercoralis* co-infection [110]. A meta-analysis of 101 studies in this area confirmed that helminth infections negatively impact vaccine outcomes [111]. Another study found that the treatment of children in Gabon with albendazole, an antihelminthic, improved their immune response to influenza vaccination [112]. Some authors have expressed concern that helminth infections may impact the effectiveness of vaccines for COVID-19 and might necessitate additional vaccine doses, though we were unable to find any studies specifically examining the impact of Strongyloides infection on the COVID-19 vaccine immunogenicity or efficacy [113]. The identification of participants with Strongyloides co-infections should be prioritized in future COVID-19 vaccine research, as this area clearly warrants further investigation. 

It has also been suggested that cross-reactivity with parasitic infections could lower the specificity of COVID-19 serology testing. A study investigated the impact of helminth infections on the specificity of three different SARS-CoV-2 antibody detection assays, which were tested on 559 serum samples collected before July 2019 and known to be seropositive for several parasitic infections [114]. Although the authors found that the specificity of the 50 sera samples with Strongyloides seropositivity was preserved, there was a lower specificity found in the specimens seropositive for visceral leishmaniasis and African trypanosomiasis. 

## 8. Discussion of the Challenges in Prevention, Diagnosis, and Management of Strongyloides Hyperinfection and COVID-19

Strongyloidiasis remains an understudied and neglected tropical disease, with the highest burden of disease that impacts LMIC settings. As the COVID-19 pandemic has led to an increased use of corticosteroids to treat patients in need of respiratory support and placed a huge strain on healthcare resources, our imperfect and outdated understanding of Strongyloides has not prepared providers with reliable estimates of the pre-test probability of Strongyloides infection. Contemporary knowledge about Strongyloides prevalence is lacking in both endemic and non-endemic areas, in both LMIC and HIC settings, which is expected to negatively impact the performance of the proposed screening algorithms for chronic strongyloidiasis risk prior to the treatment of COVID-19 with corticosteroids. The combination of this unknown prevalence of Strongyloides and limited diagnostic strategies make weighing the risks and benefits of using corticosteroids for COVID-19 a difficult task. We summarize the gaps in our current knowledge and practice with regards to COVID-19 and Strongyloides in Table 2. 

In LMIC settings, the diagnosis of chronic strongyloidiasis was already challenging prior to the COVID-19 pandemic, with serology being unable to distinguish between current and prior treated infections in higher endemicity settings, and alternative diagnostic methods being time-consuming and more technically difficult (e.g., Baermann concentration or agar plate culture techniques), or resource-intensive and unavailable (PCR). Implementing screening algorithms for Strongyloides prior to the use of corticosteroids for COVID-19 in LMIC settings with a moderate to high Strongyloides prevalence is likely to be very challenging. The more pragmatic approach of giving 1–2 doses of ivermectin for the presumptive treatment of Strongyloides prior to corticosteroid administration, while likely to be well-tolerated by patients, has healthcare resource implications that need to be studied. Furthermore, the COVID-19 pandemic may have diverted these limited healthcare resources (such as laboratory personnel, equipment, or reagents) away from non-COVID-19-related activities in these settings.

In HIC settings, strongyloidiasis is predominantly a disease found in immigrants, migrants, and refugees. It also affects populations living in poor, rural areas in these same settings, such as the southern United States. These historically underserved populations may face barriers to healthcare access which could otherwise preemptively address their chronic underlying medical issues. Providers working in non-endemic areas remain unfamiliar with tropical infectious diseases, and this deficit is magnified by the limited access that foreign-born populations have to medical care. The longstanding, asymptomatic chronic phase of strongyloidiasis may make even remote exposures and travel history relevant, and an overburdened health system may not reliably detect such important clues from a patient’s history, especially when language and cultural barriers are also present. 

In the setting of Strongyloides and COVID-19, machine learning models could be utilized to predict and evaluate the population-level impact of screening algorithms or empiric ivermectin treatment in settings of differing (or uncertain) Strongyloides prevalence, with a view to optimize clinical approaches. However, diagnostic uncertainty due to the lack of a gold standard ‘test-of-cure’ for Strongyloides represents a challenge in this context. Machine learning approaches have been used in the immunomodulatory impact of helminth infections in animal models of autoimmune diseases, and could potentially be used to study the impact of Strongyloides co-infection on COVID-19 severity and outcomes [115]. However, some authors have noted challenges with the application of COVID-19 disease models, which were mostly developed in HIC settings, to support the policy decisions in LMIC settings, and recommend collaborative approaches which would also likely benefit the models addressing Strongyloides and COVID-19 [116].

In addition to improving our knowledge of the distribution and prevalence of Strongyloides, clinical studies are needed to establish the performance and validity of the proposed screening and treatment algorithms of Strongyloides, and to investigate the impact of Strongyloides infection on the immune responses to both the COVID-19 illness and vaccinations. Strongyloides seroprevalence surveys and their validation of screening algorithms could be incorporated into the protocols for COVID-19 therapeutic and vaccine trials in efficient and cost-effective ways (e.g., by using serum already collected to study the SARS-CoV-2 antibody responses for a vaccine clinical trial), which would also help to reduce the confounding of the immunologic and clinical outcomes of COVID-19 research by Strongyloides infection. Prospective clinical trials to establish safe and effective approaches for Strongyloides treatment in persons who cannot take oral medications are also needed, as well as in special populations such as pregnant women and young children where adequate safety and pharmacokinetic data is lacking [101]. There is also a critical need for improved Strongyloides diagnostic testing that can be implemented in resource-limited settings; if the PCR capacity has increased in some LMIC settings due to the need to test for SARS-CoV-2, this technical knowledge and equipment could perhaps be leveraged in the future to perform stool PCR-based screening for soil-transmitted helminths as well. Table 3 provides specific recommendations for the future research needed in this area. 

## 9. Conclusions

The use of corticosteroids for the treatment of severe COVID-19 has placed many more people around the world at risk for Strongyloides hyperinfection, and has underscored the current deficiencies in our epidemiologic knowledge of and diagnostic testing methods for Strongyloides infection. The case reports of Strongyloides hyperinfection in the setting of COVID-19 summarized in this review likely represent a vast underreporting of cases, particularly in the higher endemicity areas. The currently proposed strategies for screening and treatment of Strongyloides infection, prior to corticosteroid initiation, require prospective clinical validation particularly in settings with limited resources so that they can be feasible and sustainable. An area of particular focus should be the comparison between Strongyloides screening versus preventive treatment with ivermectin. Finally, the interplay between Strongyloides and COVID-19 also provides opportunities to accelerate research that would address these deficiencies, partly by leveraging research networks that are conducting studies on COVID-19 treatment and vaccination. 

## Figures and Tables

**Table 1 tropicalmed-08-00127-t001:** Key demographic and clinical characteristics of the previously published cases of Strongyloidiasis in persons with COVID-19.

Reference	Patient Age (Years) and Sex	Reporting Country	Patient’s Country of Birth	Time Since Immigration (Years)	COVID-19 Treatment	Strongyloides Diagnostic Methods Used	Estimated Interval Between COVID-19 Symptom Onset and Strongyloides Diagnosis (Days)	Diagnosis	Strongyloides Treatment	Outcome
Kim [15]	63 M	United States (California)	Cambodia	NS	Dexamethasone, barcitinib	Stool exam negative; BAL fluid microscopy positive; serology positive	28	SHS	Ivermectin	Died
Gautam [16]	53 M	India	India	NA	Methylprednisolone	Stool exam positive	60	SHS	Ivermectin and albendazole	Survived
Lier [17]	68 M	United States (Connecticut)	Ecuador	20	Methylprednisolone, tocilizumab	Stool exam negative; sputum sample positive	27	SHS	Ivermectin and albendazole	Survived
Núñez-Gómez [18]	45 M	Spain	Ecuador	20	Dexamethasone	Stool exam positive; Serology positive	12	SHS	Ivermectin	Survived
O’Dowling [19]	60 F	Ireland	Nigeria	22	None	Serology positive; pathology of small bowel resected with parasites	NA (asymptomatic SARS-CoV-2)	SHS	Ivermectin	Survived
Babazadeh [20]	70 M	Iran	Iran	NA	Dexamethasone	Histopathology of gastric and duodenal biopsy	21	SHS	Ivermectin and albendazole	Survived
Patel [21]	72 M	United States	Nicaragua	NS	Dexamethasone	Stool exam positive; BAL fluid microscopy	NS	SHS	Ivermectin	Survived
Marchese [22]	59 F	Northern Italy	Southern Italy	NS	Dexamethasone, tocilizumab	Stool exam positive; serology positive	32	Chronic strongyloidiasis	Ivermectin	Survived
Feria [23]	44 M	Spain	Bolivia	17	Dexamethasone	Serology positive	7	Chronic strongyloidiasis	Ivermectin	Survived
Feria [23]	74 F	Spain	Honduras	7	Dexamethasone	Serology positive	10	Chronic strongyloidiasis	Ivermectin	Survived
Pintos-Pascual [24]	70 M	Spain	Ecuador	12	Methylprednisolone, tocilizumab, anakinra	Stool exam positive; serology positive	55	Chronic strongyloidiasis	Ivermectin and albendazole	Survived
Alkaabba [25]	76 M	United States	United States	NA	Dexamethasone	Stool exam positive; histopathology of duodenum	14	Chronic strongyloidiasis	Ivermectin	Survived
Busaidi [26]	55 M	Oman	Oman	NA	Dexamethasone	Stool exam positive	30	Chronic strongyloidiasis	Ivermectin and albendazole	Survived
Stylemans [27]	59 M	Belgium	Ecuador	7	Methylprednisolone, anakinra	PCR on stool exam positive; serology positive	60	Chronic strongyloidiasis	Ivermectin	Survived
Singh [28]	58 M	India	India	NA	Methylprednisolone	Stool exam positive	6	Chronic strongyloidiasis	Ivermectin and albendazole	Survived
Lorenzo [29]	37 F	Spain	Bolivia	NS	Steroids	Serology positive	NS	Chronic strongyloidiasis	NS	Survived
Lorenzo [29]	47 F	Spain	Bolivia	NS	Steroids	Serology positive	NS	Chronic strongyloidiasis	NS	Survived
Lorenzo [29]	33 F	Spain	Honduras	NS	Steroids	Serology positive	NS	Chronic strongyloidiasis	NS	Survived
Lorenzo [29]	38 M	Spain	Honduras	NS	No steroids or tocilizumab	Serology positive	NS	Chronic strongyloidiasis	NS	Survived
Lorenzo [29]	22 M	Spain	Morocco	NS	No steroids or tocilizumab ^	Serology positive	NS	Chronic strongyloidiasis	NS	Survived
Lorenzo [29]	69 F	Spain	Columbia	NS	No steroids or tocilizumab	Serology positive	NS	Chronic strongyloidiasis	NS	Survived
Lorenzo [29]	27 F	Spain	Peru	NS	No steroids or tocilizumab	Serology positive	NS	Chronic strongyloidiasis	NS	Survived

M = male, F = female, NS = not specified, NA = not applicable. ^ This patient did not receive corticosteroids for COVID-19 but was receiving infliximab, adalimumab, corticosteroids, and azathioprine for Crohn’s disease.

**Table 2 tropicalmed-08-00127-t002:** Gaps in current knowledge and practice.

Domain	Deficit
Immunology	Impact of Strongyloides on COVID-19 disease severity and prognosis. Impact of infection with SARS-CoV-2 on immunologic responses to parasitic infections including Strongyloides. Impact of chronic Strongyloides infection on the immunologic response to COVID-19 vaccination.
Epidemiology and public health	Limited data regarding geographic distribution and prevalence of chronic Strongyloides infection in both low endemicity and higher endemicity settings. Imprecision of Strongyloides prevalence estimates due to methodologic differences in how Strongyloides was diagnosed in previous research studies. Lack of epidemiologic and mathematical models that address the combined impact of Strongyloides and COVID-19 in different geographic settings and populations.
Screening at risk populations	Lack of awareness of Strongyloides among healthcare providers in low endemicity settings which would lead to missed screening opportunities. Lack of clinical validation of proposed screening algorithms for both low endemicity/high endemicity settings. Need for screening algorithms that are pragmatic and capable of being implemented in both HICs and LMICs.
Diagnostic testing	Lack of data regarding the sensitivity of serologic testing for Strongyloides in persons with COVID-19 (with/without the use of corticosteroids/immunosuppression). Limited understanding of how to utilize PCR testing for Strongyloides diagnosis and screening in settings where this is feasible, and the need to identify alternative diagnostic strategies (e.g., Baermann method), which could give timely results in resource-limited settings where PCR and/or serologic testing are not available. Lack of a commercially available inexpensive diagnostic test that is highly sensitive and can be used to very rapidly and effectively rule out chronic Strongyloides infection for screening prior to corticosteroid use.
Treatment	Currently there are no approved and evidence-based alternatives to ivermectin for persons unable to take oral medications, including many of those critically ill with COVID-19.
Research study settings	Inconsistent Strongyloides screening of participants in clinical trials of COVID-19 therapeutics. Overlapping symptoms and signs of worsening COVID-19 and SHS could act as a confounder in clinical trials in higher endemicity settings where the investigational treatment for COVID-19 has immune suppressive or antiparasitic effects.
Other special populations	Relative lack of safety data for ivermectin use in infants, and those who are breastfeeding or pregnant, the latter being a group at higher risk for complications of COVID-19. Difficulties addressing the disproportionate burden of Strongyloides infection in populations with additional barriers to accessing medical care, such as migrants and refugees.

Abbreviations: COVID-19, coronavirus disease 2019; SARS-CoV-2; severe acute respiratory syndrome coronavirus 2; HIC, high-income country; LMIC, low- and middle-income country; PCR, polymerase chain reaction; and SHS, Strongyloides hyperinfection.

**Table 3 tropicalmed-08-00127-t003:** Recommendations for future research.

Domain	Future Research Needed
Immunology	In vitro and in vivo studies to understand how Strongyloides modulates immune response to SARS-CoV-2 and whether outcomes and severity of COVID-19 are different in co-infected individuals. In vitro and in vivo studies to understand how COVID-19 modulates immune responses in chronic Strongyloides infection and whether this predisposes or protects individuals from SHS. Prospective studies examining the impact of chronic Strongyloides infection on B- and T-cell-mediated immune responses to COVID-19 vaccination in higher endemicity settings.
Epidemiology and public health	Strongyloides seroprevalence surveys in low endemicity settings to better define geographic distribution of strongyloidiasis and inform screening programs. Epidemiologic studies and mathematical modeling to assess the combined impact of Strongyloides and COVID-19 within communities and to assist governments and public health bodies in identification of successful mitigation strategies. Seroprevalence surveys for Strongyloides IgG among persons who have been diagnosed with COVID-19, ideally in combination with demographic and social data such as travel and occupation to identify subgroups of patients with COVID-19 at highest risk for subclinical Strongyloides infection.
Screening at risk populations	Development of evidence-based educational interventions for healthcare providers in low endemicity settings to increase knowledge, awareness, and reduce missed Strongyloides screening opportunities. Multi-site validation of proposed screening algorithms for both low, moderate, and high endemicity settings. For high endemicity settings with limited resources, comparison of the health and economic outcomes of Strongyloides screening algorithms versus empiric treatment with 1–2 doses of ivermectin prior to initiation of corticosteroids.
Diagnostic testing	Sensitivity/specificity estimates from studies examining the performance in persons with COVID-19 of different serologic testing assays used for Strongyloides screening prior to corticosteroid initiation. Development of an inexpensive diagnostic test (such as a rapid antigen test) that is highly sensitive, requires little technical ability/training, and can be used to very rapidly effectively rule out chronic Strongyloides infection for screening prior to corticosteroid use. Increased availability and implementation of Strongyloides PCR assays that can be used in LMIC settings, preferably using the same testing platforms as PCR assays currently being used for SARS-CoV-2 diagnosis in these settings. Prospective evaluation to optimize alternative diagnostic methods that have a short turnaround time (such as Baermann method) for use in resource-limited settings where PCR is unavailable, e.g., for screening for asymptomatic Strongyloides infection in high endemicity settings prior to corticosteroid administration.
Treatment	Pharmacokinetic studies to establish to optimal dosing and safety outcomes of subcutaneous formulations of ivermectin. Prospective clinical trials of subcutaneous ivermectin use for chronic strongyloidiasis and SHS to establish its safety and efficacy for use in critically ill patients with COVID-19 when the oral route cannot be used.
Research settings	Development and implementation of screening protocols for subclinical Strongyloides infection in both low and higher endemicity settings for participants in clinical trials of COVID-19 therapeutics. Seroprevalence surveys, testing for Strongyloides IgG by ELISA, for example, could be performed on stored serum specimens that may have been previously collected for COVID-19 research studies. Reporting of strongyloidiasis (including SHS and disseminated Strongyloides) and other parasitic infections, as an adverse event in studies of COVID-19 therapeutics, and as an adverse event of special interest where the investigational drug is expected to have immune modulatory or antiparasitic effects.
Special populations	Prospective studies are needed to confirm the safety and appropriate dosing for ivermectin in special populations such as infants, and those who are breastfeeding or pregnant. Health policy interventions are needed to ensure that underserved populations such as refugees, migrants, or incarcerated persons can access screening for Strongyloides if they are diagnosed with COVID-19 and require corticosteroids.

Abbreviations: SARS-CoV-2; severe acute respiratory syndrome coronavirus 2; COVID-19, coronavirus disease 2019; LMIC, low- and middle-income country; SHS, Strongyloides hyperinfection; and ELISA, enzyme-linked immunosorbent assay.

## Data Availability

Not applicable.

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
