# Peer review of "Strongyloides and COVID-19: Challenges and Opportunities for Future Research"

_tropicalmed, 2023, doi:10.3390/tropicalmed8020127_

Round 1

Reviewer 1 Report

General comments

An interesting and complete perspective on Strongyloides and COVID-19 that offers an attractive big picture about that very neglected tropical disease.

Many of the high endemic countries are also suffering from very limited resources. The Baermann method is not discussed even though it has a comparable performance and is the recommended exam for those countries due to its ease of use and implementation (cf. specific comments).

In my point of view, PCR test is presented as a unique option is this article, especially as a proposal for Low Income Countries where this method is not an option.

I would suggest adapting the diagnostic recommendations so that their applicability is not limited to only high-income countries.

Specific comments

#1 Paragraph starting Line 241.

If historically direct stool exam has been used, the reference exam is the Baermann method for quite a while. Its performance is comparable to the PCR.

Recent microbiological studies on the field in Bolivia and Lao (LMIC) concluded that Baermann test is the exam of choice (Getaz 2019 doi: 10.1371/journal.pntd.0007028, Chankongsin 2020 doi: 10.1186/s40249-020-00750-y).

Due to the ease of implementation in LMIC, after comparing it, Chankongsin recommended Baermann test + Koga plate over PCR in low- and middle-income settings in Southeast Asia.

Meurs and al. conclude that although being slightly more sensitive, PCR is in general not feasible in low-resource settings (Meurs 2017, doi: 10.1371/journal.pntd.0005310)

#2 Paragraph starting line 470

This paragraph is about vaccine immune response and helminth infection, but the article is specifically about Strongyloides stercoralis and COVID-19. I did not find any mention of S. stercoralis in the provided sources. Therefore, I do not find this paragraph relevant. I suggest to find solid specific references regarding that subject.

#3 Table 2 “Diagnostic testing”

“Limited understanding of how best to utilize PCR testing for Strongyloides diagnosis and screening, and how best to implement this in settings with limited resources or low ende-micity”

 I suggest that you emphasize the importance of systematic implementation of the Baermann method in resource-limited countries where PCR is not yet available.

Author Response

Thank you for your insightful and detailed comments and for taking the time to review our manuscript.

  1. On line 254 and 537 of the paper the Baermann method is discussed with mention of the reliance on this method in LMIC.  On line 257 we discuss PCR as an option used in some studies but note that it is not widely available at this time and may be difficult to implement in LMIC with limited laboratory resources. In response to the reviewer comments we have added the reference by Getaz et al 2019 (doi: 10.1371/journal.pntd.0007028) to highlight the successful implementation of the Baermann method in an endemic area and have also added the reference by Meurs et al 2017 noting the lack of feasibility of PCR in many LMIC settings. 
  2. During our extensive literature search, we were unable to find any specific studies into the impact of Strongyloides coinfection on and COVID-19 vaccine immune responses and therefore were not able to include any specific references with regard to Strongyloides and COVID-19 vaccines, though we do cite one reference where the authors had specific concern regarding the impact of helminth coinfection of COVID-19 vaccine efficacy (Chacin-Bonilla et al 2021). In response to reviewer comments we have added an additional recent reference to a study that noted S. stercoralis coinfection negatively impacted IgG production in response to a candidate malaria vaccine (Nouatin et al 2021) and we have edited this section to more specifically focus on the need for future research studies to assess whether there is any impact of Strongyloides co-infection specifically on COVID-19 vaccine response.  In Table 3 (Recommendations for future research) we suggest additional studies to specifically understand potential modulation of the immune response to COVID-19 vaccination in persons with Strongyloides infection.  
  3. In response to the reviewer comments regarding the Baermann method, we have updated Table 2 (Gaps in Current Knowledge and Practice, diagnostic testing domain) to emphasize implementation of the Baermann method in settings where PCR is not feasible.

Reviewer 2 Report

It is an interesting paper involved with the situation of Covid-19 pandemic outbreak.

Author Response

Thank you for your thoughtful comments and for taking the time to review our manuscript.

Reviewer 3 Report

The paper focuses on the identification of challenges unique to screening, diagnosis, and treatment of strongyloidiasis in the context of COVID-19. The paper also highlights the knowledge gaps and proposes a number of specific strategies that can be addressed in the future. The paper reviews the related and recent literature extensively, specifies the challenges and opportunities. The paper can be improved further by considering the points below:

1)      Abstract of the paper should be improved. The first sentence can state the importance of the content, then the gaps in the corresponding literature. Key contributions of the paper should be expressed clearly and then the major findings of the paper should be provided.

2)      Introduction has provided some background researches and highlighted their advantages and disadvantages. However, critical review of the recent and related works are not quite strong. The corresponding gaps should be emphasized strongly and based on these gaps, the claimed contributions of the paper should be justified.

3)      The paper can express the internal and external uncertainties affecting the outcomes of the diagnosis and treatments.

4)      The paper stated that “A recent study using a spatiotemporal statistical modelling approach estimated a global prevalence of 8.1%, corresponding to 613.9 million people infected worldwide”. Mathematical modelling, machine learning and artificial intelligence approaches are widely considered to analyse, predict and manipulate future treatments under various constraints. As an efficient future tools, this paper can dedicate a sub-section on modelling prediction models. I would suggest these recent and related papers: Development of a multi-dimensional parametric model with non-pharmacological policies for predicting the COVID-19 pandemic casualties. Pharmacological, non-pharmacological policies and mutation: An artificial intelligence based multi-dimensional policy making algorithm for controlling the pandemic diseases. The first paper develops a mathematical model which is enriched with the external models of the non-pharmacological policies. The model managed to predict the second peak of the COVID-19. The last paper generates 5-dimensional policies to control the future pandemic casualties under the limited vaccination and various uncertainties.

5)      Paragraph main sections and sub-sections of the paper should be reorganized to enhance the presentation of the work. Some of the sub-sections seem quite long.

Good luck with the improvements…

Author Response

Thank you for your detailed and thoughtful comments and for taking the time to review our manuscript.  

  1. We have rewritten and improved our abstract in line with the reviewer comments and included the major findings of the review within the abstract. 
  2. The introduction has been revised in response to the reviewer comments. 
  3. With regard to the internal and external uncertainties regarding the outcomes of the diagnoses and treatments within the previous cases of Strongyloidiasis in the setting of COVID-19, we note the suspected publication bias regarding the low mortality rate of the previously published case reports and the scarcity of reports from higher endemicity settings such as Latin America. With regard to the internal and external uncertainties regarding the diagnosis of Strongyloides in the setting of COVID-19 in general, our article highlights the lack of a gold standard diagnostic test and the relative merits and limitations of different diagnostic testing and screening strategies, with particular attention to the limitations of serology and the lack of feasibility of some diagnostic and screening tools within low and middle income country settings. We hope that the revisions to our paper including to Table 2, which details specific gaps in our knowledge and understanding, more clearly outline these uncertainties and limitations. 
  4. We read with interest several additional references suggested by the reviewer highlighting the benefits of mathematical modeling and machine learning to predict COVID-19 burden in different settings and carefully considered the potential application of these models to the possible impact of Strongyloides on COVID-19 outcomes in different geographic settings. In response to the articles suggested by the reviewer we have highlighted this topic as an additional current knowledge gap in Table 2 and as a key area for future research which we have integrated into Table 3, however, we felt that a more detailed review of this topic was beyond the scope of the current review article.  
  5. We have added additional subheadings to break up some of the longer sections in our review in response to the reviewer comments and to improve readability. We would like to thank the reviewer for drawing our attention to this. 

Round 2

Reviewer 3 Report

The revised paper has been re-assessed. Even though the main manuscript shows the deleted and added parts, since the authors did not state each concern raised in the first round of the revision and clearly expressed what has been done to address specific concern, the assessment of the revised paper was challenging as well. From the previous report:

Comment 1: The abstract of the paper is shortened and some new sentences are added. However, the related gaps in the current literature and how this paper is filling such gaps are not strong enough. The title has “challenges and opportunities” , therefore, the paper should reflect such properties. In addition, “future research” should be expanded since it can refer to a large number of research areas which are not considered yet.

Comment 2: Introduction is unfortunately still raw. It should analyse the recent and related works with the advantages and disadvantages. Critical review is important for such a paper as it aims to reveal theoretical and practical requirements in the future.

Comment 3: Please use academic language to express the uncertainties. They can be internal or external. They can be random or they can have a character and all of these play a different role on the outcomes. There are recent papers in the literature which address complex and multi-dimensional uncertainties.

Comment 4: The authors expressed the improvement, but they are not clearly available and understandable in the paper. Please emphasize the Mathematica modelling, machine learning and artificial intelligence approaches. 

Comment 5: It seems that the paragraphs and main sections of the paper have been improved as requested.

Author Response

Thank you for taking the additional time to review our manuscript.  

Comment 1: The abstract of the paper is shortened and some new sentences are added. However, the related gaps in the current literature and how this paper is filling such gaps are not strong enough. The title has “challenges and opportunities” , therefore, the paper should reflect such properties. In addition, “future research” should be expanded since it can refer to a large number of research areas which are not considered yet.

  • We appreciate your detailed  feedback on our revised abstract.  We have added an additional detail about future research that is needed, as suggested by the reviewer. At the same time, we are mindful of the journal's 200 word limit for the abstract and therefore space constraints prevent us from expanding this section further as our abstract now comprises 218 words after the aforementioned revision.

Comment 2: Introduction is unfortunately still raw. It should analyse the recent and related works with the advantages and disadvantages. Critical review is important for such a paper as it aims to reveal theoretical and practical requirements in the future.

  • We have reworked our introduction to provide a more detailed review of Strongyloides, but we are also conscious of  trying to avoid overlapping too much in content with the recent (and excellently written) general review of Strongyloides recently published by this journal in October 2022 by Luvura et al. We feel that the introduction now satisfactorily establishes a background of Strongyloidiasis and hyperinfection syndrome related to steroid use for COVID-19 in preparation for later parts of the manuscript which provide a critical review of additional related works on the topic.

Comment 3: Please use academic language to express the uncertainties. They can be internal or external. They can be random or they can have a character and all of these play a different role on the outcomes. There are recent papers in the literature which address complex and multi-dimensional uncertainties.

  • We appreciate the reviewers detailed feedback about discussion of uncertainties that would affect models and outcomes and have added an additional paragraph at the end of section 3.1 on epidemiologic risk to detail uncertainties as requested and informed by the  framework for uncertainty reporting developed for soil transmitted helminth modeling reported by Araujo Navas et al (2017) which we have included as a citation. 

Comment 4: The authors expressed the improvement, but they are not clearly available and understandable in the paper. Please emphasize the Mathematica modelling, machine learning and artificial intelligence approaches.  

  • In response to the detailed reviewer feedback requesting additional discussion of mathematical modeling, machine learning and artificial intelligence approaches, we have added two additional sections to our manuscript to meet this request (one at the end of section 3.1, the other in the penultimate paragraph of section 8). We were unable to find any examples of these approaches being used to address Strongyloides in the specific setting of COVID-19 but have added additional references for the two examples we found of these approaches being applied to Strongyloides (Barratt and Sapp 2020, Rubel et al 2020) and we discuss some potential applications of these approaches to Strongyloides in the setting of COVID-19 and also some challenges (including uncertainties as outlined above in response to number 3.). We feel that more detailed discussion of the subject of Covid-19 pandemic modeling in general (outside of the context of Strongyloides) is beyond the scope of this paper and also beyond the expertise of the authors, noting this area has been the focus of  meta-analyses and review articles in its own right (see Teerawattonanon 2022, which we have added as a citation, as one example).

Comment 5: It seems that the paragraphs and main sections of the paper have been improved as requested.

  • Thank you for the helpful suggestions regarding our subsections. We are pleased with the revisions we made (as suggested by the reviewer) to improve clarity and readability. 

Round 3

Reviewer 3 Report

The paper has been revised and can be accepted with the current version.